# The Functional Role of Group 2 Innate Lymphoid Cells in Asthma

**DOI:** 10.3390/biom13060893

**Published:** 2023-05-26

**Authors:** Takahiro Matsuyama, Kentaro Machida, Keiko Mizuno, Hiromi Matsuyama, Yoichi Dotake, Masahiro Shinmura, Koichi Takagi, Hiromasa Inoue

**Affiliations:** Department of Pulmonary Medicine, Graduate School of Medical and Dental Sciences, Kagoshima University, Kagoshima 890-8520, Japan

**Keywords:** airway inflammation, asthma, comorbidity, group 2 innate lymphoid cells

## Abstract

Asthma is a heterogeneous disease characterized by chronic airway inflammation. Group 2 innate lymphoid cells (ILC2) play an important role in the pathogenesis of asthma. ILC2s lack antigen-specific receptors and respond to epithelial-derived cytokines, leading to the induction of airway eosinophilic inflammation in an antigen-independent manner. Additionally, ILC2s might be involved in the mechanism of steroid resistance. Numerous studies in both mice and humans have shown that ILC2s induce airway inflammation through inflammatory signals, including cytokines and other mediators derived from immune or non-immune cells. ILC2s and T helper type 2 (Th2) cells collaborate through direct and indirect interactions to organize type 2 immune responses. Interestingly, the frequencies or numbers of ILC2 are increased in the blood and bronchoalveolar lavage fluid of asthma patients, and the numbers of ILC2s in the blood and sputum of severe asthmatics are significantly larger than those of mild asthmatics. These findings may contribute to the regulation of the immune response in asthma. This review article highlights our current understanding of the functional role of ILC2s in asthma.

## 1. Introduction

Asthma is a heterogeneous disease, characterized by chronic airway inflammation, variable expiratory airflow limitation, and a history of respiratory symptoms, such as wheezing, shortness of breath, chest tightness, and cough, which vary in intensity and over time [1]. It affects approximately 300 million people worldwide, with the prevalence increasing in developed countries.

Recent clinical and translational research has demonstrated that asthma is a heterogeneous disease comprising various phenotypes and endotypes. In terms of phenotype, asthma encompasses eosinophilic asthma, neutrophilic asthma, mixed granulocytic asthma, and pauci-granulocytic asthma [2]. Mixed granulocytic asthma is a phenotype characterized by increased levels of both eosinophils and neutrophils. Pauci-granulocytic asthma is a phenotype characterized by normal levels of both eosinophils and neutrophils [3]. Among these phenotypes of asthma, eosinophilic asthma, characterized by eosinophilia in the airways or blood driven by type 2 immune responses, can be induced by allergic and non-allergic mechanisms, mainly involving T helper type 2 (Th2) cells and group 2 innate lymphoid cells (ILC2), respectively.

Over the past decade, ILCs have been identified as a component of the innate immune system that can interact with various hematopoietic and non-hematopoietic cells to coordinate immunity, inflammation, and homeostasis in multiple organs throughout the body [4]. Unlike T cells and B cells, ILCs lack antigen-specific receptors and lineage (Lin) markers, and they cause antigen-non-specific immune responses [5,6]. ILCs consist of three subsets—ILC1s, ILC2s, and ILC3s—which are characterized by their transcription factors and the cytokines that they produce. These subsets correspond to Th1, Th2, and Th17 cells, respectively. Among these ILCs, ILC2s have been discovered in the gut, spleen, liver, and bone marrow [5,7,8]. ILC2s produce a large amount of IL-5 and IL-13 in response to epithelial-cell-derived cytokines, such as IL-25, IL-33, and thymic stromal lymphopoietin (TSLP), which are considered to play an essential role in the pathogenesis of allergic disorders, including asthma.

In this review, we highlight our current understanding of the functional role of ILC2s in asthma, especially focusing on eosinophilic asthma.

## 2. Airway Inflammation in Asthma Pathogenesis

The characteristics of asthma are chronic airway inflammation and airway hyperresponsiveness (AHR). Airway inflammation involves various inflammatory cells, such as eosinophils, neutrophils, lymphocytes, and mast cells, as well as airway structural cells including airway epithelial cells, fibroblasts, and airway smooth muscle cells, along with various humoral factors [9]. Persistent airway inflammation causes airway remodeling, which leads to irreversible airflow limitations. Furthermore, AHR is considered to be primarily induced by airway inflammation, whereas AHR is induced even when airway inflammation is mild with airway remodeling, because repeated bronchoconstriction promotes airway remodeling [10].

The airways of asthmatic patients exhibit pathologies, such as goblet cell metaplasia, excessive subepithelial collagen deposition, airway smooth muscle hyperplasia, and increased vascularity [11]. These findings are considered to be caused by inflammatory mediators, including cytokines and chemokines, produced by inflammatory cells and airway structural cells (Figure 1).

Dendritic cells (DCs) phagocytose antigens and present them to naïve CD4^+^ T cells, which differentiate into Th2 cells via the surrounding cytokine and produce type 2 cytokines, such as IL-4, IL-5, and IL-13. IL-4 induces the expression of vascular cell adhesion molecule-1 (VCAM-1) in vascular endothelial cells, which leads to eosinophil aggregation. IL-4 also induces the isotype class switching of B-cells to IgE synthesis [12]. Antigen-specific IgE produced by B cells activates mast cells, which release inflammatory mediators such as histamine and leukotrienes (LTs), triggering an immediate asthmatic response [12]. IL-4 plays a pivotal role in Th2 cell differentiation. IL-13, which shares IL-4 receptor α (IL-4Rα), has many functions that overlap with those of IL-4 [13]. IL-13 induces airway smooth muscle contraction, mucus hypersecretion, and inducible nitric oxide synthase (iNOS) in bronchial epithelial cells, resulting in an increased level of fractional exhaled nitric oxide (FeNO), in addition to the IL-4-mediated production of IgE and the increased expression of VCAM-1 in vascular endothelial cells [13,14]. IL-5 promotes the proliferation, maturation, activation, and survival of eosinophils and the translocation of eosinophils from the bone marrow into the systemic circulation [13,15]. These cytokines play a crucial role in the induction of allergic airway inflammation and the pathogenesis of asthma. Moreover, naïve CD4^+^ T cells differentiate into Th17 cells in the presence of IL-6, IL-21, and TGF-β [16]. Th17 cells produce IL-17A, IL-17F, and IL-22 and contribute to the defense against extracellular parasites and autoimmune diseases. However, their involvement in asthma has also been demonstrated. In a murine model of asthma, the adoptive transfer of antigen-specific Th17 cells mediated airway neutrophilic inflammation and AHR, which promoted steroid resistance [17]. Thus, IL-17 is involved in severe asthma, neutrophilic asthma, asthma exacerbations, and airway remodeling involving the recruitment of neutrophils [18]. However, anti-IL-17 biologics provided no clinical benefit in asthmatics [19]. Therefore, it is necessary to design clinical trials with neutrophilic asthmatics.

ILC2s provide host defense against helminth infection, contributing to inflammatory responses and tissue repair. ILC2s depend on transcription factor GATA3 for their development and function, and they produce significant amounts of IL-5 and IL-13 in response to IL-25, IL-33, and TSLP, leading to type 2 immune responses. In addition, since ILC2s induce type 2 immune responses in Rag1^−/−^ mice lacking T cells and B cells, ILC2s have been shown to produce type 2 cytokines in a T-cell-independent manner [20]. Thus, ILC2s play an essential role in innate-immunity-mediated type 2 airway inflammation. ILC3s require transcription factor RORγ for their induction and produce IL-17A, IL-17F, and IL-22 [21]. The ILC3-mediated production of IL-17, as well as Th17 cells, may contribute to asthma, especially neutrophilic asthma.

## 3. Interaction of ILC2 with Other Immune Cells

ILC2s interact with various effector cells to orchestrate type 2 immune responses as part of the complicated immune system for allergic reactions (Figure 2).

### 3.1. ILC2s and T Cells

The functional relationship between ILC2s and T cells has been reported. IL-2 derived from T cells proliferates ILC2s and produces ILC2-derived IL-13 [22,23]. In addition to IL-2, Th2-cell-derived IL-4 and IL-13 have also been shown to promote ILC2 proliferation in the lungs of helminth-infected mice using the genetic deletion of IL-4/IL-13 [24]. On the other hand, IL-13 produced by ILC2 is involved in Th2 cell differentiation by inducing the migration of DCs to the draining lymph nodes [25]. ILC2-derived IL-9 also activates DCs to promote the differentiation of Th2 cells [26]. Furthermore, some ILC2s express MHC class II to act as antigen-presenting cells, leading to enhanced Th2 cell differentiation [23,27]. These findings demonstrate that ILC2s and T cells activate each other to promote airway inflammation through cytokine secretion or cell-to-cell contact.

### 3.2. ILC2s and DCs

As mentioned above, IL-9 and IL-13 secreted by ILC2s induce DC activation and migration to the draining lymph nodes, respectively, leading to the differentiation of CD4^+^ T cells into Th2 cells [25,26]. Tumor-necrosis-factor-like protein 1A (TL1A), which is derived from DCs and macrophages, regulates the adaptive immune response by co-stimulating T cells [28]. TL1A is a member of the tumor necrosis factor (TNF) superfamily and is expressed in non-immune cells, such as synovial fibroblasts and endothelial cells, as well as immune cells. TL1A binds to its receptor, death receptor 3 (DR3), and contributes to the induction of innate and adaptive immune homeostasis [29]. Moreover, TL1A promoted pulmonary ILC2 function via DR3 in a papain-induced murine model [30]. In an *Alternaria*-induced murine model, IFN-α production by activated plasmacytoid DCs (pDCs), a subtype of DCs, inhibited ILC2 proliferation and increased their apoptosis rate, leading to alleviated airway inflammation and AHR. A study using human pDCs and ILC2s found that toll-like receptor 7 (TLR7) agonist-activated pDCs produced IFN-γ, leading to the significantly reduced production of IL-5 and IL-13 derived from ILC2s [31]. These results suggest that DCs might provide new insights into the function of ILC2s.

### 3.3. ILC2s and Macrophages

Macrophages, divided into M1 and M2 macrophages, produce type 1 and type 2 cytokines, respectively. M2 macrophages, in particular, play an important role in the pathogenesis of asthma. ILC2s activated by IL-2 or IL-33 produce IL-13, which in turn induces M2 macrophages to clear helminths in the lungs [32]. In a study using mice infected with rhinovirus on days 6 and 13 of life, the lungs of M2-macrophage-lacking mice infected with rhinovirus showed decreased production of epithelial-derived cytokines, decreased numbers of ILC2s, and decreased mRNA expression of IL-5 and IL-13. Thus, M2 macrophages may be involved in ILC2-driven airway inflammation and mucous metaplasia during early-life rhinovirus infections [33]. However, it has not been clarified how M2 macrophages affect the airway epithelium to increase IL-33 expression. Alveolar macrophages have been shown to produce IL-33 itself when mice are infected with influenza, leading to IL-13 production derived from ILC2s [34]. In contrast, interstitial macrophages produce IL-27 with IL-33 stimulation, which has a suppressive effect on ILC2 activation [35]. These findings suggest that macrophages could have both an activating and an inhibitory effect on ILC2s.

### 3.4. ILC2s and Mast Cells

Mast cells play a crucial role in IgE-dependent acute allergic responses by releasing lipid mediators. Mast cells also interact with ILC2s in an IgE-independent manner to develop airway inflammation. In fact, mast cells produce prostaglandin D_2_ (PGD_2_), which activates ILC2-mediated airway inflammation [36]. Additionally, mast-cell-derived IL-2 increases the number of regulatory T cells (Tregs) and promotes IL-10 production from Tregs, thereby suppressing papain- or IL-33-induced airway eosinophilic inflammation [37]. Mast cells are also activated by IL-9 released from IL-33-elicited ILC2s, which induces IL-2 production. However, previous studies found that helminth-infected wild-type mice showed the comparable accumulation of mast cells in the lungs as compared to IL-9-receptor-deficient mice [38], indicating that the effect of IL-9 on mast cells is ambiguous.

### 3.5. ILC2s and Basophils

Basophils have phenotypic similarities to mast cells and have been erroneously identified as mast cells [39]. Recently, there has been increasing evidence to suggest that basophils play an essential role in allergic responses. In fact, individuals with asthma, especially those with eosinophilic asthma, exhibit increased frequencies of sputum basophils. Furthermore, the frequencies of sputum basophils are positively correlated not only with sputum eosinophils but also with blood eosinophils and FeNO levels [40,41]. Basophils, along with other types of immune cells, also interact with ILC2s to orchestrate airway inflammation. IL-4 production from basophils, in combination with IL-33, enhances ILC2-derived type 2 cytokine production, subsequently leading to papain-induced airway eosinophilic inflammation [42]. A similar mechanism has also been observed in skin [43]. Our group has demonstrated that a long-acting muscarinic receptor antagonist, which is used for asthma, suppresses basophil-derived IL-4 production and subsequently regulates ILC2 activation to ameliorate airway eosinophilic inflammation in a murine model [44]. Therefore, basophils might have a supportive effect by inducing airway inflammation.

## 4. Modulators of ILC2 Function in Asthma

Recent studies have revealed that ILC2s express multiple receptors on their surfaces and that various factors enhance or repress ILC2 activation and cytokine secretion upon airway inflammation [45]. Specifically, in addition to cytokines, lipid mediators, neuropeptides, neurotransmitters, and hormones influence the activation or suppression of ILC2s (Table 1).

### 4.1. Cytokines

In addition to IL-33, IL-25, and TSLP, other cytokines also have critical roles in regulating ILC2 activation. IL-2 promotes cell survival and the proliferation of ILC2s and functions as a co-factor for ILC2-derived type 2 cytokine production to induce airway inflammation [46,47]. As described above, IL-4 in conjunction with IL-33 activates ILC2 to secrete type 2 cytokines, leading to eosinophil proliferation and mucin production [42]. Additionally, human ILC2s express the IL-4 receptor, and IL-4 in synergy with IL-33 promotes ILC2 proliferation and type 2 cytokine production [48]. Although IL-4 cannot be secreted by ILC2s in response to IL-33 stimulation, recent studies have demonstrated that the administration of PGD_2_ and leukotriene D_4_ (LTD_4_) induces IL-4 production from ILC2s [49,50,51]. ILC2-derived IL-4 might act on ILC2s as an autocrine factor by binding to an IL-4 receptor [52]. In the context of human helminthiasis, ILC2-derived IL-4 has been shown to contribute to ILC2 expansion and Th2 cell differentiation [51]. However, this finding has only been demonstrated in the small intestine and has not been reported in the lungs. ILC2-derived IL-9 induces airway eosinophilic inflammation by activating ILC2s themselves in an autocrine manner during helminth infection [38], indicating that IL-9 is also crucial for ILC2 function and survival.

On the other hand, some cytokines inhibit ILC2 function. IL-27 suppressed the function of tissue-resident ILC2s in mice exposed to papain and *Alternaria* [53,54]. A recent study revealed that a TLR7 agonist stimulated IL-33-induced interstitial macrophages to produce IL-27, suppressing ILC2-driven airway inflammation [35]. Among interferons, interferon-α (IFN-α), which is derived from plasmacytoid DCs, as well as IFN-β and IFN-γ, has been shown to inhibit ILC2 activation and regulate airway inflammation [31,53,55]. IFN-α and IFN-β attenuated type 2 cytokine production from human ILC2s in vitro [31,55]. A TLR3 agonist-induced IFN-β has been shown to suppress *Alternaria*-induced airway inflammation by inhibiting the effects of STAT5-activating cytokines, such as IL-2, IL-7, and TSLP, on lung ILC2s [56].

### 4.2. Lipid Mediators

Lipid mediators—such as LTs and PGs—play a critical role in regulating the development or resolution of airway inflammation by modulating the balance between pro-inflammatory and anti-inflammatory mediators. LTs and PGs are generated by arachidonic acid metabolism, and these mediators are released from various cells, including immune cells.

LTs mainly consist of two groups, LTB_4_ and the cysteinyl (Cys) LTs—LTC_4_, LTD_4_, and LTE_4_. In studies of murine lungs, the CysLT receptor (CysLTR) was expressed on ILC2s. Among CysLTs, LTD_4_ produces not only IL-5 and IL-13 but also IL-4 from ILC2s in vitro. Furthermore, LTD_4_ enhances *Alternaria*-induced airway eosinophilic inflammation [49]. LTC_4_ and LTE_4_ have also been shown to induce high levels of IL-5 and IL-13 production derived from ILC2s stimulated with IL-33 in vitro. In an in vivo study, LTC_4_ also potentiated further IL-33-driven airway eosinophilic inflammation [57]. Moreover, the treatment of montelukast, a leukotriene receptor antagonist, suppresses ILC2-mediated airway eosinophilic inflammation via CysLT [49]. CysLTR has two different types: CysLT1R and CysLT2. LTC_4_ binds equally to CysLT1R and CysLT2R, whereas LTD_4_ binds preferentially to CysLT1R. In a murine model, the synergistic effect of LTC_4_ with IL-33 on airway eosinophilic inflammation was completely suppressed in CysLT1R^−/−^ mice, but not CysLT2R^−/−^ mice [57]. On the other hand, as for LTB_4_, in a study of a papain-induced murine model, genetic deletion or an antagonist of BLT1, a high-affinity receptor for LTB_4_, resulted in decreased papain-induced IL-33 expression derived from the most likely airway epithelial cells, leading to suppressed ILC2 activation and the decreased migration of DCs [58]. These findings suggest that the LTB_4_–BLT1 axis may indirectly mediate the activation of ILC2.

PGs associated with ILC2s have been shown to be PGD_2_, PGE_2_, and PGI_2_. The chemoattractant receptor-homologous molecule expressed on Th2 cells (CRTH2), a PGD_2_ receptor, is expressed in the ILC2s of mice and humans. PGD_2_ treatment increases ILC2 numbers in the lungs in vivo. Furthermore, CRTH2-deficient mice had diminished ILC2-mediated pulmonary inflammation in a helminth-induced murine model [59]. Thus, PGD_2_ acts as a positive regulator for ILC2s. Murine ILC2s express E-type prostanoid receptor (EP) 1 and EP4, which are PGE_2_ receptors. Treatment with PGE_2_ resulted in the reduced production of IL-5 and IL-13, which led to attenuated airway eosinophilic inflammation in an IL-33-driven murine model [60]. In a study using human ILC2s, PGE_2_ also inhibited the ILC2-derived IL-5 and IL-13 production and ILC2 proliferation. Furthermore, while PGE_2_ primarily acts on murine ILC2s through EP4, both EP2 and EP4 are associated with the action of PGE_2_ on human ILC2 [61]. IP, a PGI_2_ receptor, is also expressed in both human and murine ILC2s. In an *Alternaria*-induced murine model, treatment with a PGI_2_ analog led to a reduced number of lung IL-5- and IL-13-expressing ILC2s. In an in vitro study of human ILC2s that were stimulated by IL-2 and IL-33, a PGI_2_ analog also inhibited the production of IL-5 and IL-13 [62]. These findings indicate that PGE_2_ and PGI_2_ have a direct inhibitory effect on ILC2-mediated airway inflammation and may potentially be used for asthma treatment.

### 4.3. Neuropeptides and Neurotransmitters

Neuropeptides and neurotransmitters are released by neurons to act on various tissues and cells, including immune cells. Neuropeptides are involved in the respiratory tract and affect ILC2 function. Neuromedin U (NMU) has been found to be a pro-inflammatory driver in type 2 immune responses. NMU is expressed in the thoracic dorsal root ganglia, including afferent sensory neurons. NMUR1, a receptor for NMU, is expressed on ILC2s among immune cells. NMU activates ILC2s, leading to the facilitation of ILC2-induced type 2 airway inflammation in a murine model [63]. Furthermore, NMU was shown to activate ILC2s during a viral respiratory infection [64]. In a human study, NMU increased the levels of IL-5 and IL-13 derived from human ILC2s in a dose-dependent manner and induced cell migration, including eosinophils [65]. Calcitonin-gene-related peptide (CGRP) is released from pulmonary neuroendocrine cells (PNECs) that reside near ILC2s at airway branchpoints [66]. CGRP receptors, CALCRL and RAMP1, are expressed on ILC2s. PNEC-derived CGRP induces the increased production of IL-5 from ILC2s, which leads to airway eosinophilic inflammation [66]. On the other hand, other reports have revealed that the *Il5*^hi^ subpopulation of ILC2s expresses *Calca*-encoding CGRP in addition to CGRP receptors [67], and CGRP administration suppressed IL-33 production or papain-induced pulmonary eosinophilic inflammation and worm expulsion in a murine model [67,68]. These conflicting findings suggest that CGRP may potentially act on lung ILC2s in an autocrine and/or paracrine manner.

In regard to neurotransmitters, epinephrine is secreted by sympathetic nerves, including adrenergic neurons. Murine and human ILC2s express β_2_-adrenergic receptor (β_2_ AR), to which epinephrine binds. In a study using β_2_ AR-deficient mice or β_2_ AR agonist treatment, β_2_ AR contributed to the inhibitory effect on ILC2-dependent airway eosinophilic inflammation [69]. Acetylcholine (ACh) is released by the parasympathetic nerves and binds to muscarinic and nicotinic receptors to act on various cells. ILC2s express α7-nicotinic ACh receptor (α7nAChR). Treatment with a α7nAChR agonist ameliorated ILC2-dependent airway eosinophilic inflammation and AHR in an *Alternaria*-induced murine model. Further, the same inhibitory effects of an α7nAChR agonist on airway inflammation were observed in a humanized mice model [70]. In contrast, as for muscarinic receptors, our group reported that the muscarinic 1–3 receptors were not expressed on murine lung ILC2s in western blotting [44]. However, qPCR has shown that the muscarinic 4 receptor is expressed on lung ILC2s. In fact, in a helminth-infected murine model, ACh treatment led to further heightened ILC2 responses, such as higher numbers of total, IL-5-, and IL-13-producing ILC2s in the lungs, leading to airway eosinophilic inflammation. Moreover, ILC2s express choline acetyltransferase (ChAT), which catalyzes the synthesis of ACh. Thus, these results suggest that the ChAT–ACh pathway contributes to the promotion of type 2 innate immunity to helminth infection [71]. Although muscarinic 1–3 receptors are functionally recognized in the lungs, including the airways, associated with asthma pathogenesis, the association between muscarinic 4 receptor and asthma is not yet understood. Further studies are required.

### 4.4. Hormones

Females have a higher prevalence and severity of asthma as compared to men. The numbers of ILC2s in the sputum of patients with mild allergic asthma are greater in females, as well as those in the blood of asthmatics [72,73]. These phenomena suggest a role for sex hormones as regulators of asthma pathogenesis.

Sex hormones consist of androgen and estrogen. Androgen deprivation by orchiectomy abolished the sex differences in ILC2 development and ameliorated IL-33-mediated lung inflammation in a murine model [74]. Testosterone has also been shown to attenuate *Alternaria*-induced airway eosinophilic inflammation by reducing the expression of IL-33 and TSLP and the number of ILC2s in the lungs [73]. Androgen receptors are expressed on ILC2 progenitors (ILC2P), and these findings suggest that androgen signaling has a suppressive role in ILC2-dependent airway eosinophilic inflammation. On the other hand, estrogen receptors are not expressed on ILC2s. However, estrogen receptor-α signaling exacerbated ILC2-mediated airway inflammation by increasing the release of IL-33 from airway epithelial cells in an *Alternaria*-induced murine model [75]. These findings suggest that estrogen acts on ILC2s indirectly through airway epithelial cells and that ovarian hormone fluctuations may be involved in the peri-menstrual worsening of asthma in women.

Glucagon-like peptide-1 (GLP-1) is an incretin hormone secreted by the intestines and the central nervous system, regulating metabolic, cardiovascular, and neuroprotective functions [76]. Already, the GLP-1 receptor (GLP-1R) agonist has been used against obesity and type 2 diabetes. In humans, GLP-1R exhibits high expression in the lungs, especially lung epithelial cells [77,78]. Asthmatics prescribed GLP-1R agonists for type 2 diabetes have been shown to have lower frequencies of asthma exacerbation [76]. In studies using *Alternaria*-induced mice, GLP-1R agonists significantly suppressed ILC2-mediated airway eosinophilic inflammation by reducing IL-33 production derived from airway epithelial cells [77]. Further, TSLP levels in bronchoalveolar lavage fluid (BALF) were significantly higher in *Alternaria*-induced obese mice compared to lean mice, and the elevated levels of BALF TSLP in obese mice were significantly reduced by GLP-1R agonists, suppressing ILC2-driven airway eosinophilic inflammation [79]. These findings suggest that the GLP-1R agonist might be a potential therapeutic agent for asthma, especially asthma associated with obesity.

## 5. Impact and Functional Role of ILC2 in Asthma

### 5.1. Impact of ILC2 in Asthma

In several studies involving humans, the contribution of ILC2s to asthma pathogenesis has been reported. Asthma patients have been found to exhibit increased frequencies and numbers of ILC2s in their peripheral blood [80,81,82,83]. The frequencies of ILC2s in BALF are also increased in asthmatic patients compared to healthy individuals [84]. Sputum analysis has also shown that allergen exposure can induce increased numbers of ILC2s in asthmatics [85]. The frequencies of ILC2s positively correlate with disease severity and airway eosinophilic inflammation—such as sputum and peripheral blood eosinophils and fractional exhaled nitric oxide (FeNO)—and negatively correlate with predicted FEV1% [80]. Significantly more ILC2s are detected in the peripheral blood and sputum of severe asthmatics as compared with mild asthmatics [86]. Additionally, IL-5- and IL-13-expressing ILC2 numbers are also increased in severe asthmatics [83,86].

On the other hand, ILC3s increased in the lungs of obese mice fed a high-fat diet, and they developed AHR. Moreover, the development of ILC3s producing IL-17 in these obese mice was dependent on the NLRP3 inflammasome. In human studies, IL-17-producing ILC3s have been identified in the BALF of asthmatics, particularly in patients with severe asthma, where the ILC3s producing IL-17 were more numerous than in those with mild asthma or non-asthma [87]. ILC3s may contribute to neutrophilic asthma via IL-17, leading to obesity-associated asthma. However, it remains unclear whether IL-17-producing ILC3 numbers are increased in obese asthmatics, and reports on ILC3s in asthma remain limited.

As described above, the prevalence and severity of asthma are greater in females than in males. This may indicate that fluctuations in sex hormone levels during the menstrual cycle and pregnancy are associated with asthma pathogenesis in females [88]. In animal models, female mice have been found to exhibit more ILC2s in the lungs than male mice [73,74]. In a study of patients with moderate-to-severe asthma, women showed significantly more ILC2s in their peripheral blood than men [73]. These findings indicate that ILC2s might contribute to the pathogenesis and severity of asthma, and that sex hormones might also play a role in the sex differences in ILC2 numbers.

### 5.2. Functional Role of ILC2s in Asthma Pathogenesis

In a murine model of asthma, protease allergens—such as papain and house dust mites (HDM)—and *Aspergillus fumigatus* damaged airway epithelial cells, releasing epithelial-derived cytokines such as IL-25, IL-33, and TSLP. These cytokines stimulate ILC2s to produce large amounts of IL-5 and IL-13 from ILC2s. However, a recent study using single-cell RNA sequencing to compare tissue-resident ILC2s found that the IL-33 receptor was highly expressed in the lungs, adipose tissue, and bone marrow, while the IL-25 receptor was highly expressed in the intestinal tract [89]. Thus, among epithelial-derived cytokines, IL-33 and TSLP might contribute to ILC2-associated asthma. The results from murine experiments have been applied to human asthmatics to further understand asthma’s pathogenesis.

#### 5.2.1. Steroid Resistance

In an ovalbumin (OVA)+IL-33-induced murine model, the production of IL-5 and IL-13 from ILC2s was more resistant to treatment with dexamethasone than that from CD_4_^+^ T cells. This study shows that TSLP is associated with steroid resistance in pulmonary ILC2s by activating Bcl-xL, an anti-apoptotic molecule, through the STAT5 pathway [90]. In human studies, TSLP expression in the airway epithelium and lamina propria was increased in patients with severe asthma [91]. In addition, ILC2s in BALF from asthmatic patients were resistant to steroids; furthermore, BALF from asthmatic patients had elevated TSLP levels, and the TSLP levels in BALF positively correlated with the steroid resistance of ILC2s [92]. A recent study has also shown that there is another signaling pathway that induces the steroid resistance of ILC2s. ILC2s in the sputum of asthmatics exhibit high expression of DR3 upon allergen exposure or cytokine stimulation, such as IL-2, IL-33, and TSLP. The levels of TL1A, which is the ligand for DR3, in the sputum of asthmatic patients are also significantly increased by allergen exposure, and they are higher in patients with prednisolone-dependent severe eosinophilic asthma than in those with mild asthma. The TL1A-induced activation of ILC2s is not inhibited by dexamethasone when stimulated with TSLP. Thus, these findings suggest that the TL1A–DR3 axis in the presence of TSLP induces steroid-resistant ILC2s [93]. Human circulating ILC2s express CD45RA, which is converted to CD45RO in response to stimulation, such as alarmins. In fact, there is a large population of CD45RO-expressing ILC2s in the nasal polyps (NPs) of patients with chronic rhinosinusitis with NPs (CRSwNPs) [94]. These ILC2s correlate with disease severity and resistance to corticosteroid therapy. On the other hand, once tissue-resident murine ILC2s are activated after helminth or fungal infection, they have different characteristics, called inflammatory ILC2s (iILC2s); they tend to migrate and stay at the inflammatory site and produce type 2 cytokines continually for a prolonged period [53]. The transcriptomic features of CD45RO-expressing ILC2s in NPs are similar to those of murine iILC2s [94]. Thus, ILC2s in BALF and sputum may have characteristics similar to those of murine iILC2s and may express CD45RO.

#### 5.2.2. Viral-Induced Asthma Exacerbation

Viral infection is considered to be a major cause of asthma exacerbation [95]. Viral infection promotes an innate immune response by inducing the production of IFN and pro-inflammatory cytokines such as IL-1α, IL-1β, IL-6, IL-8, and TNF-α. IL-8, released from epithelial cells, contributes to the accumulation of neutrophils at the inflammatory site, inducing airway neutrophilic inflammation. However, viral infection can also induce the recruitment and activation of eosinophils in the respiratory tract [95]. ILC2s are associated with innate immunity because ILC2s cannot respond to specific antigens as with T cells but can induce airway eosinophilic inflammation. Therefore, ILC2s might play an important role in virus-induced innate immune responses and asthma exacerbation.

Respiratory syncytial virus (RSV), rhinovirus (RV), and influenza virus are representative respiratory viruses. RSV is a major cause of infant hospitalizations, and severe RSV infections are an important risk factor for childhood asthma [96]. In a RSV-infected murine model, RSV triggered the proliferation and activation of IL-13-producing ILC2s in a TSLP-dependent manner [97]. NMU production was induced in the RSV-infected lungs, and the administration of NMU induced the proliferation and activation of pulmonary ILC2s through the NMUR1 pathway, leading to the exaggeration of RSV-induced airway inflammation [64]. RSV infection also promoted the expression of IL-33 and an increased number of ILC2s in the lungs of neonatal mice [96]. Infants with severe RSV infection had a higher increase in ILC2 numbers and higher production of IL-4, IL-13, and IL-33 in nasal fluids than those with moderate disease [98]. In regard to RV, bronchial epithelial cells from asthmatic patients inoculated with RV exhibited high expression of IL-33 mRNA and protein [99]. In another study of asthmatic patients exposed to RV, asthmatic patients showed elevated levels of IL-33 and type 2 cytokines in nasal fluids as compared with healthy subjects. These elevated cytokines correlated with the severity of asthma exacerbation. In addition, bronchial IL-33 levels also positively correlated with the bronchial levels of IL-5 and IL-13 [100]. RV infection produced IL-33 from airway epithelial cells in vitro, and human ILC2s co-cultured with RV-infected airway epithelial cells significantly produced type 2 cytokines [100]. In another study of asthmatic patients inoculated with RV, the patients had significantly increased ILC2 numbers and ratios of ILC2 to ILC1 in BALF as compared with healthy subjects. Elevated ratios of ILC2 to ILC1 in asthmatic patients positively correlated with the severity of the exacerbation and type 2 cytokine production in nasal mucosal lining fluids [101]. In a study of mice infected with the influenza A virus, influenza virus infection caused AHR through the macrophage–IL-33–ILC2–IL-13 axis, regardless of Th2 cells [34]. However, this study did not show that the influenza virus directly affected IL-33 production in airway epithelial cells.

Coronavirus disease 2019 (COVID-19), which is caused by severe acute respiratory syndrome coronavirus 2 (SARS-CoV-2), has spread rapidly around the world since it was first reported in December 2019. Asthma has not been shown to contribute to a higher risk of severe COVID-19 or a worse prognosis, and asthmatic patients are shown to have a reduced risk of mortality as compared with non-asthmatic patients [102,103]. Furthermore, among asthma patients, asthmatics with lower eosinophil levels (absolute eosinophil counts (AEC) < 150/μL)) show more frequent hospitalization and mortality due to COVID-19 as compared to those with higher eosinophil levels (AEC > 150/μL) [104,105]. Thus, in asthma patients, eosinophils exert a possible protective effect. However, in regard to the relation between COVID-19 and ILC2s, COVID-19 patients have higher levels of serum IL-33 than healthy subjects, along with an increased number of circulating ILC2s [106]. Furthermore, another study has demonstrated elevated frequencies of circulating ILC2s in patients with moderate COVID-19 as compared with healthy subjects [107]. Therefore, COVID-19 infection may induce asthma exacerbation, and further examination is required to address this discrepancy between clinical and basic studies.

## 6. Impact of ILC2 on Asthma Comorbidities

Some patients with asthma have several common comorbidities. The importance of managing comorbidities has been shown because some comorbidities contribute to poor asthma control, leading to a greater symptom burden and impaired quality of life [1]. ILC2s are also involved in the pathogenesis of these comorbid disorders, which can cause asthma to be uncontrolled or exacerbated.

### 6.1. Aspirin-Exacerbated Respiratory Disease

Aspirin-exacerbated respiratory disease (AERD) is an inflammatory condition that consists of three clinical features: asthma, rhinosinusitis with recurrent NPs, and sensitivity to non-steroidal anti-inflammatory drugs (NSAIDs), such as aspirin, that inhibit cyclooxygenase-1 (COX-1), a synthetic enzyme of PGs [108]. Therefore, lipid mediators have a critical role in AERD. Following AERD-inducing pathognomonic COX-1-inhibiting reactions, concentrations of pro-inflammatory mediator (PGD_2_ and CysLTs) rapidly increased. Furthermore, in patients with AERD, ILC2 numbers significantly increased in nasal fluids and decreased in the blood at the time of COX-1-inhibiting reactions. The frequencies of ILC2s positively correlated with the levels of urinary LTE_4_ and PGD_2_ and symptom severity [109]. These findings suggest that the lipid mediator–ILC2 axis may contribute to AERD pathogenesis.

### 6.2. Chronic Rhinosinusitis with Nasal Polyps

CRSwNP is one subgroup of CRS that is characterized by the development of polyps in the lining of the nose and paranasal sinuses. It is characterized by moderate-to-severe type 2 inflammation with hypereosinophilia and elevated IgE concentrations [110]. The local mucosa of CRSwNP patients exhibits increased numbers of ILC2s, and ILC2 are even more numerous in CRSwNP patients with comorbid asthma [111]. Furthermore, as described above, the frequencies of CD45RO-expressing ILC2 are increased in the NPs of CRSwNP patients and in the blood of patients with CRS or asthma, which positively correlates with disease severity and steroid resistance [94]. This particular subtype of ILC2s in NPs may spread into the bloodstream and contribute to the pathogenesis of asthma. This mechanism may explain why CRSwNP patients are more likely to have comorbid asthma.

### 6.3. Allergic Rhinitis

Allergic rhinitis is an inflammatory disorder induced by an IgE-mediated reaction with increasing numbers of Th2 cells and type 2 cytokines in the nasal mucosa [112]. This disease is common, affecting 55–60% of asthmatics. Regarding the relationship between allergic rhinitis and ILC2s, a nasal allergen challenge in patients with allergic rhinitis induced an increased number of blood ILC2s, indicating active ILC2 recruitment to the circulation [113]. Furthermore, ILC2 numbers were found to be increased through IL-33 derived from myeloid DCs [114]. On the other hand, another study indicated that the numbers of ILC2s in the peripheral blood were not remarkably altered in patients with allergic rhinitis [82]. Moreover, in a study using ragweed-sensitized T/B-cell-deficient Rag2^−/−^ mice, the frequencies of sneezing and eosinophilic infiltration were significantly decreased, suggesting that T cells play an essential role in the pathogenesis of allergic rhinitis, greater than that of ILC2s [115]. Further research is needed to understand the effect of ILC2s on allergic rhinitis.

### 6.4. Eosinophilic Granulomatosis with Polyangiitis

Eosinophilic granulomatosis with polyangiitis (EGPA) is a multisystemic disease characterized by late-onset asthma, blood and tissue eosinophilia, and small-to-medium-vessel vasculitis [116]. In patients with EGPA, ILC2 numbers in the peripheral blood at onset are significantly higher than in those with asthma, EGPA at relapse, or at remission. Moreover, in EGPA patients, the number of eosinophils in the peripheral blood is positively correlated with serum TSLP levels and peripheral blood ILC2 numbers [117]. Therefore, increased ILC2 numbers in the peripheral blood may have an influence on disease activity in EGPA. However, few studies have examined the association between EGPA and ILC2, and further research is required to better understand this relationship.

## 7. Conclusions

This review provides an overview of current understandings regarding the role of ILC2s in type 2 airway inflammation and the pathogenesis of asthma. Asthma is a heterogeneous and complex disease characterized by airway inflammation. Eosinophilic asthma is the predominant phenotype in severe asthma [118], and ILC2s have been shown to play a critical role in the recruitment of eosinophils. Moreover, the interplay between ILC2s and various factors—including cytokines, neuropeptides, and other immune cells, such as T cells—has been identified as orchestrating asthma pathogenesis. Furthermore, ILC2s are also implicated in other allergic comorbidities of asthma, which makes it crucial to address these disorders for optimal asthma management.

Currently, to achieve asthma control and to reduce the risk of exacerbations, various types of biologics have been used. Among biologics, dupilumab and tezepelumab target IL-4Rα and TSLP, respectively, and IL-4 and TSLP acted on ILC2s directly in in vivo or in vitro studies; therefore, these biologics might attenuate ILC2-mediated asthma. In a study using sorted peripheral blood ILC2s in vitro, the expression of type 2 cytokine mRNA was shown to be significantly decreased in asthmatics treated with dupilumab [119]. Thus, biologics targeting cytokines might improve our understanding of the relationship between ILC2s and asthma pathogenesis. Moreover, biologics targeting IL-33 and its receptors, which are itepekimab and astegolimab, respectively, have also shown clinical benefits in asthma treatment [120,121]. Blocking cytokines with biologics is expected to improve our understanding of the relationship between ILC2s and asthma pathogenesis. However, there are no data indicating whether these drugs act on ILC2s directly as well as tezepelumab. Therefore, further investigation is required to elucidate the precise mechanisms by which these biologics modulate ILC2-mediated asthma.

## Figures and Tables

**Figure 1 biomolecules-13-00893-f001:**
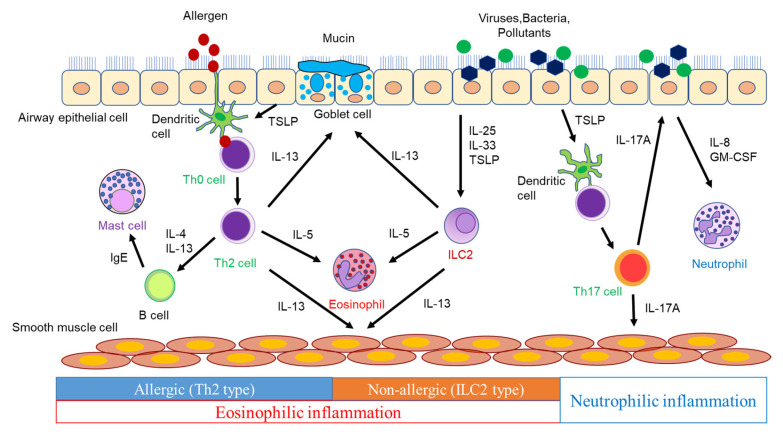
The pathogenesis of airway inflammation in asthma.

**Figure 2 biomolecules-13-00893-f002:**
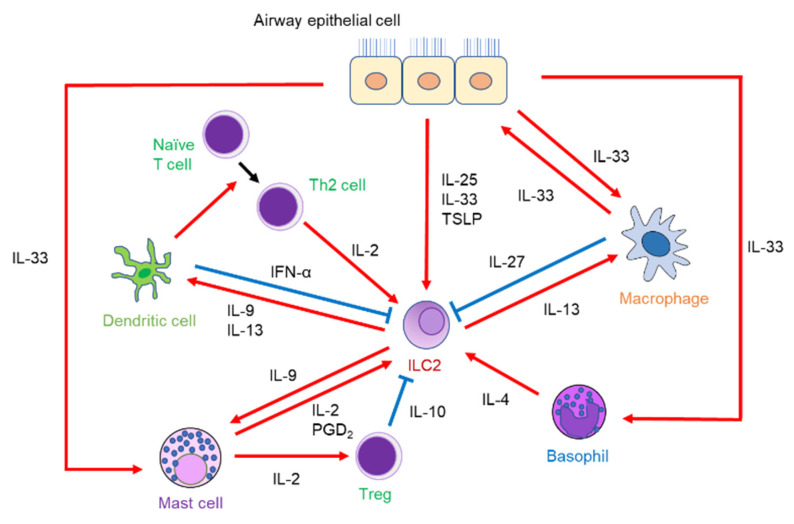
The network between ILC2s and other immune cells in the pathogenesis of airway inflammation.

**Table 1 biomolecules-13-00893-t001:** Direct regulatory factors of ILC2s.

	Ligand	Receptor		Ligand	Receptor
Activation			Inhibition		
Epithelial-derived cytokine	IL-25	IL-25R	Cytokines	IL-10	IL-10R
	IL-33	IL-33R (T1/ST2)		IL-27	IL-27R
	TSLP	TSLPR		IFN-α	IFN-α/βR
Co-stimulatory cytokines	IL-2	IL-2R		IFN-β	IFN-α/βR
	IL-4	IL-4R		IFN-γ	IFN-γR
	IL-7	IL-7R			
	IL-9	IL-9R			
	TL1A	DR3			
Lipid mediators	CysLTs (LTC_4_, LTD_4_, LTE_4_)	CysLTR	Lipid mediators	PGE_2_	EP
	LTB_4_	BLT1		PGI_2_	IP
	PGD_2_	CRTH_2_		LXA_4_	ALX/FPR_2_
Neuropeptide Neurotransmitter	NMU	NMUR1	Neuropeptide Neurotransmitter	VIP	VPAC2
	CGRP	CALCRL/RAMP1		Nicotine	α7nAChR
	ACh	Chrm4		Adrenaline	β_2_ AR
Hormones			Hormones	Androgen	Androgen R

AR, adrenergic receptor; LXA_4_, lipoxin A_4_; VIP, vasoactive intestinal peptide.

## Data Availability

Not applicable.

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
