# Peer review of "The Functional Role of Group 2 Innate Lymphoid Cells in Asthma"

_biomolecules, 2023, doi:10.3390/biom13060893_

Round 1

Reviewer 1 Report

General comments:

The article is a comprehensive review of the role of ILC2s in asthma. Since the title of the article suggests a focus on ILC2s and type 2 asthma, the text on ILC3s, IL-17 etc seems to exceed the principal aim of the review. Perhaps the title of the article needs to be changed? In some sections major statements are made without proper referencing.

Specific comments

Lines 29- 32. If the authors are generally introducing phenotypes of asthma, a brief mechanistic explanation for granulocytic and pauci-granulocytic should be given. Or neutrophilic asthma should be removed since the authors mention “especially focusing on eosinophilic asthma” (line 49). In addition, neutrophilic asthma does not involve ILC2s. The authors could introduce all phenotypes and then narrow down to T2 asthma which the authors do in later sections when discussing interactions between ILC2s and various cells.

Lines 57-58: AHR can be induced in presence of low or no inflammation. Since the article mentions pauci-granulocytic asthma in its introduction passage, it would be best to say inflammation can induce AHR but not in all instances.

Line 62, Figure 1. The figure suggests that DCs are producing IL-17. However, this does not seem to be the intent of the illustration.

Line 64. “presented to them…”.

Line 75. References are needed for several of these statements.

Line 85. The claim that IL-17 is involved in severe asthma needs to less definitive since the anti-IL-17 biologics trials have been negative, perhaps because IL-17 high asthmatic subjects have not been specifically studied. 

Line 88. Transcription factor.

Section 3:

Line 128. This statement is vague. In what way will DCs provide insights into the function of ILC2s.

Line 135. It is not obvious to the reader how studying mice lacking M2 macrophages could implicate M2 macrophages. Please explain.

Line 139: Alveolar macrophages have been shown to produce IL-33 itself when infected with influenza (Chang et al., 2011).

Line 140. ”which has a suppressive effect…”.

Line 148. “activates ILC2-mediated …”.

Line178. Modulators.

Line 194. Is this the intent of this sentence? As an IL-4 receptor”?. Do you mean “on”?  

Line 207. Should read “attenuate”.

Line 220. This sentence as written is not correct. Please insert ILC2s in the sentence.

Line 238. “Treatment with

Section 5: Impact of ILC2 in asthma

Why mention ILC3s in this section when the review is about the role of ILC2 in asthma?

Section 6: Functional role of ILC2s in asthma pathogenesis

This section aims to translate mechanistic observations to explain pathogenesis in human asthmatics. The previous section (Impact of ILC2 in asthma) could be incorporated in this section.

Line 381. This sentence needs to be re-written.

Section 8: Conclusions

It would be appropriate to dedicate a section of biologics against ILC2s with more detail than currently provided.

Minor corrections proposed in the review. 

Author Response

Response to Reviewer1

General comments:

The article is a comprehensive review of the role of ILC2s in asthma. Since the title of the article suggests a focus on ILC2s and type 2 asthma, the text on ILC3s, IL-17 etc seems to exceed the principal aim of the review. Perhaps the title of the article needs to be changed? In some sections major statements are made without proper referencing.

RESPONSE: We thank the reviewer for important comments on our manuscript.

Specific comments

1) Lines 29- 32. If the authors are generally introducing phenotypes of asthma, a brief mechanistic explanation for granulocytic and pauci-granulocytic should be given. Or neutrophilic asthma should be removed since the authors mention “especially focusing on eosinophilic asthma” (line 49). In addition, neutrophilic asthma does not involve ILC2s. The authors could introduce all phenotypes and then narrow down to T2 asthma which the authors do in later sections when discussing interactions between ILC2s and various cells.

Response: Mixed granulocytic asthma is the phenotype of asthma characterized by increased levels of both eosinophils and neutrophils. Pauci-granulocytic asthma is the phenotype of asthma characterized by normal levels of both eosinophils and neutrophils (J Asthma. 2009). We added the mechanistic insight about each of these phenotypes in the introduction section (Line 32-35).

2) Lines 57-58: AHR can be induced in presence of low or no inflammation. Since the article mentions pauci-granulocytic asthma in its introduction passage, it would be best to say inflammation can induce AHR but not in all instances.

Response: AHR is induced even when airway inflammation is mild with airway remodeling, because repeated bronchoconstriction promotes airway remodeling (N Engl J Med. 2011). We added this description in section 2 (Line 61-63).

3) Line 62, Figure 1. The figure suggests that DCs are producing IL-17. However, this does not seem to be the intent of the illustration.

Response: We changed the position of the DCs to show that Th17 cells produce “IL-17” clearly.

4) Line 64. “presented to them…”.

Response: As the reviewer indicated, “presented to them” is correct. We corrected it in Line 69.

5) Line 75. References are needed for several of these statements.

Response: We added three references for these sentences.

6) Line 85. The claim that IL-17 is involved in severe asthma needs to less definitive since the anti-IL-17 biologics trials have been negative, perhaps because IL-17 high asthmatic subjects have not been specifically studied.

Response: As the reviewer’s comment, the anti-IL-17 biologics provided no clinical benefit in asthmatics (Am J Respir Crit Care Med. 2013). Therefore, it is required to design clinical trials in neutrophilic asthmatics. We added these sentences in Line 91-93.

7) Line 88. Transcription factor.

Response: Thank you. We modified “transcription factors” to “transcription factor” in Line 95.

Section 3:

8) Line 128. This statement is vague. In what way will DCs provide insights into the function of ILC2s.

Response: A study using human pDCs and ILC2s found that toll-like receptor 7 (TLR7) agonist-activated pDCs produces IFN-γ, leading to significant reduced production of IL-5 and IL-13 derived from ILC2s (J Allergy Clin. Immunol. 2018). We corrected this description on Line 136-138.

9) Line 135. It is not obvious to the reader how studying mice lacking M2 macrophages could implicate M2 macrophages. Please explain.

Response: In the study using mice infected with rhinovirus on day 6 and 13 of life, the lungs of M2 macrophage-lacking mice infected with rhinovirus showed decreased production of the epithelial-derived cytokines, decreased numbers of ILC2s, and decreased mRNA expression of IL-5 and IL-13. We changed these descriptions in Line 145-148.

10) Line 139: Alveolar macrophages have been shown to produce IL-33 itself when infected with influenza (Chang et al., 2011).

Response: As the reviewer suggested, alveolar macrophages have been shown to produce IL-33 itself when mice are infected with influenza, leading to IL-13 production derived from ILC2s (Nat Immunol. 2011). We added these descriptions in Line 151-153.

11) Line 140. ”which has a suppressive effect…”.

Response: I corrected the word which the reviewer pointed out in Line 154.

12) Line 148. “activates ILC2-mediated …”.

Response: I corrected “ILC2s-” to “ILC2-” in Line 162.

13) Line178. Modulators.

Response: I modified “Modulator” to “Modulators” in Line 192.

14) Line 194. Is this the intent of this sentence? As an IL-4 receptor”?. Do you mean “on”? 

Response: As the reviewer’s suggestion, ILC2-derived IL-4 acts on ILC2s by binding to an IL-4 receptor. Therefore, we changed “acting as” to “binding to” to avoid “act on” repeatedly in Line 208.

15) Line 207. Should read “attenuate”.

Response: As the reviewer indicated, “attenuate” is correct. Thus, we corrected that in Line 221.

16) Line 220. This sentence as written is not correct. Please insert ILC2s in the sentence.

Response: As the reviewer commented, LTD4 produce IL-4 “from ILC2s.” We added this sentence to “from ILC2s” in Line 234. Thank you!

17) Line 238. “Treatment with

Response: We changed “of” to “with” in Line 256.

18) Section 5: Impact of ILC2 in asthma

Response: We changed “on” to “in” in Line 347.

19) Why mention ILC3s in this section when the review is about the role of ILC2 in asthma?

Response: We introduce that ILCs consist of three subsets; ILC1s, ILC2s, and ILC3s in the introduction session. As the reviewer’s comment, this review focuses on the role of ILC2s in asthma. However, because ILC3s has been shown to contribute to asthma pathophysiology, especially neutrophilic asthma, we also mentioned the role of ILC3s in the pathophysiology of neutrophilic asthma.

20) Section 6: Functional role of ILC2s in asthma pathogenesis

This section aims to translate mechanistic observations to explain pathogenesis in human asthmatics. The previous section (Impact of ILC2 in asthma) could be incorporated in this section.

Response: We reviewed the influence of ILC2s in asthma in section 5 and the role of ILC2s in asthma pathophysiology in section 6. Thus, as the reviewer recommended, we merged section 6 into section 5.

21) Line 381. This sentence needs to be re-written.

Response: We re-wrote this sentence as follows: the TL1A-induced activation of ILC2s is not inhibited by dexamethasone when stimulated with TSLP (Line 405-407).

22) Section 8: Conclusions

It would be appropriate to dedicate a section of biologics against ILC2s with more detail than currently provided.

Response: The reviewer’s comment is very important. In this review, we described that anti-IL-33 antibody (itepekimab) and anti-ST2 antibody (astegolimab) provided clinical benefits in asthma. However, there has been no data about whether these drugs act on ILC2s directly as well as Tezepelumab. Thus, we considered that it was difficult to dedicate the section of these drugs. We added the following sentence in Line 552-553: However, there has been no data about whether these drugs act on ILC2s directly as well as tezepelumab.

Reviewer 2 Report

This review article well summarizes roles of ILC2 in asthma. Clinically and basically important findings to data are included enough. The reviewer would like to raise some comments, which may be useful to improve this article.

1) Lines 215-255: It is better to distinguish CysLT1 and CysLT2 receptors.

2) Lines 280-288: This paragraph on ACh-induced ILC2 responses does not make sense. Additional explanation should be required.

3) Lines 368-393: It has been reported that both mepolizumab (Bel, New Eng J Med, 2014) and benralizumab (Nair, New Eng J Med, 2017)exerted glucocorticoid-sparing effects. These findings suggest that IL-5 produced from ILC2 could play roles in the induction of steroid resistance. How do authors feel this consideration?

4) Lines 375-383: Because TLA1 and DR3 have not been generally known, thes molecules should be briefly explained. 

Author Response

Response to Reviwer2

This review article well summarizes roles of ILC2 in asthma. Clinically and basically important findings to data are included enough. The reviewer would like to raise some comments, which may be useful to improve this article.

Response: We thank the reviewer for the positive comments on our manuscript.

1) Lines 215-255: It is better to distinguish CysLT1 and CysLT2 receptors.

Response: The CysLT receptor (CysLTR) has two different types: CysLT1R and CysLT2. LTC4 binds equally to CysLT1R and CysLT2R, whereas LTD4 binds preferentially to CysLT1R. In a murine model, the synergistic effect of LTC4 with IL-33 on airway eosinophilic inflammation was completely suppressed in CysLT1R-/- mice, but not CysLT2R-/- mice (J Immunol. 2017). We commented these sentences in Section 4 (Line 240-244).

2) Lines 280-288: This paragraph on ACh-induced ILC2 responses does not make sense. Additional explanation should be required.

Response: In a helminth-infected mouse model, ACh treatment exhibits further heightened ILC2 responses, such as higher numbers of total, IL-5 and IL-13-producing ILC2s in lung, leading to airway eosinophilic inflammation. Moreover, ILC2s express choline acetyltransferase (ChAT), which catalyzes the synthesis of ACh. Thus, these results suggest that the ChAT-ACh pathway contributes to promoting type 2 innate immunity to helminth infection (Science Immunology. 2021). We made corrections to these statements in Line 301-306.

3) Lines 368-393: It has been reported that both mepolizumab (Bel, New Eng J Med, 2014) and benralizumab (Nair, New Eng J Med, 2017)exerted glucocorticoid-sparing effects. These findings suggest that IL-5 produced from ILC2 could play roles in the induction of steroid resistance. How do authors feel this consideration?

Response: Thank you for your important comment. As the reviewer pointed out, mepolizumab and benralizumab have an oral glucocorticoid-sparing effect in patients relying on oral glucocorticoids to manage severe asthma. We have reported that glucocorticoid can suppress only a limited number of responses induced by overexpressed IL-13 (Am J Respir Crit Care Med. 2003). In contrast, glucocorticoid suppresses cytokine production generally. In this review, we highlighted, in an OVA+IL-33-induced murine model, IL-5/IL-13 production from ILC2s is more resistant to the treatment with dexamethasone than from CD4+ T cells (Nat Commun. 2013). We added these descriptions in Line 392-393.

4) Lines 375-383: Because TLA1 and DR3 have not been generally known, these molecules should be briefly explained.

Response: TL1A is a member of TNF superfamily, and is expressed in non-immune cells, such as synovial fibroblast and endothelial cell, as well as immune cells. TL1A binds to its receptor death receptor 3 (DR3) and contributes to inducing innate and adaptive immune homeostasis (Front Immunol. 2022.). These descriptions were added in Section 2 (Line 129-132).